# pH-Responsive Liposome–Hydrogel Composite Accelerates Nasal Mucosa Wound Healing

**DOI:** 10.3390/pharmaceutics17060690

**Published:** 2025-05-24

**Authors:** Yingchao Yang, Jingyi Chen, Shengming Wang, Yaxin Zhu, Yao Wang, Yan Chen, Mingjiang Xia, Ming Yang, Hongliang Yi, Kaiming Su

**Affiliations:** 1Department of Otorhinolaryngology Head and Neck Surgery, Shanghai Key Laboratory of Sleep Disordered Breathing, Shanghai Sixth People’s Hospital Affiliated to Shanghai Jiao Tong University School of Medicine, 600 Yishan Road, Shanghai 200233, China; yangyingchao2019@sjtu.edu.cn (Y.Y.); 1377649534com@sjtu.edu.cn (J.C.); wangshengming@sjtu.edu.cn (S.W.); zhuyaxin@alumni.sjtu.edu.cn (Y.Z.); seihowy@sjtu.edu.cn (Y.W.); chen_yan@sjtu.edu.cn (Y.C.); niuniuwang@sjtu.edu.cn (M.X.); 2Otolaryngology Institute of Shanghai Jiao Tong University, 600 Yishan Road, Shanghai 200233, China; 3Department of Urology, Shanghai Sixth People’s Hospital Affiliated to Shanghai Jiao Tong University School of Medicine, 600 Yishan Road, Shanghai 200233, China

**Keywords:** nasal mucosa, pH-responsive, hydrogel, acylhydrazone bond, dexamethasone, liposome, antibacterial

## Abstract

**Objectives**: Nasal mucosa wound healing faces challenges such as acidic microenvironments and bacterial proliferation. Persistent mucosal defects predispose to complications such as nasal septal perforation. Conventional drug delivery systems suffer from nonspecific release and short-term efficacy. This study aimed to develop a pH-responsive liposome-hydrogel composite (HYD-Lip/DXMS@HG) to integrate pH-triggered dexamethasone (DXMS) delivery, antifouling properties, and mechanical support for refractory injuries. **Methods**: The composite combined acylhydrazone-modified liposomes with a hydrogel synthesized from hydroxyethylacrylamide (HEAA) and diethylacrylamide (DEAA). In vitro assays evaluated DXMS release kinetics, RPMI 2650 cell migration/proliferation, and antibacterial properties. In vivo rabbit nasal mucosal injury models assessed healing efficacy via histology analyses. RNA sequencing was performed to identify key signaling pathways. **Results**: HYD-Lip/DXMS@HG exhibited sustained DXMS release in acidic conditions, accelerating cell migration/proliferation in vitro. In rabbits, the composite reduced TNF-*α* expression and CD45+ leukocyte infiltration, while enhancing collagen alignment and epithelial thickness. RNA sequencing identified upregulated ECM receptor interaction, Hippo, TGF-*β*, and PI3K-Akt pathways, linked to collagen remodeling, anti-apoptosis, and angiogenesis. **Conclusions**: This multifunctional platform synergizes pH-triggered drug delivery, mechanical support, and antibacterial activity, offering a promising therapeutic strategy for refractory nasal mucosal injuries and postoperative recovery.

## 1. Introduction

The nasal mucosa serves as a critical barrier in the respiratory system, efficiently removing pathogens and particulate matter through the mucociliary clearance system while regulating inhaled air temperature, humidity, and initiating local immune responses [1,2]. However, nasal mucosal injuries caused by trauma, chronic inflammation, or surgery often led to persistent defects, increasing the risk of complications such as septal perforation, impaired mucociliary function, and recurrent infections, which severely affect patient quality of life [3,4,5,6,7,8]. The open physiological environment of the nasal cavity poses unique challenges for mucosal repair: continuous exposure to airborne pathogens elevates infection risks [9]; lysozymes and proteases in nasal secretions accelerate drug degradation; and the persistent motion of cilia shortens local drug retention time despite aiding in foreign particle expulsion [10]. These factors collectively hinder the natural healing of severe mucosal defects.

Current clinical treatments remain insufficient to overcome these challenges. While antibiotics and intranasal steroid sprays suppress inflammation, their nonspecific release patterns are susceptible to mucociliary clearance, necessitating frequent administration [11]. Traditional packing materials (e.g., sodium hyaluronate, gelatin sponges) temporarily cover wounds but lack stable adhesion and antimicrobial properties, increasing the risk of secondary infections during prolonged use [12]. Tissue engineering approaches, such as hydrogels, show promise by providing scaffolds for drug delivery and tissue regeneration [13]. However, existing materials like chitosan microspheres or thermosensitive hydrogels face limitations, including rapid enzymatic degradation (e.g., by chitosanases) and drug burst release [14,15].

To address these issues, we developed a composite system (HYD-Lip/DXMS@HG) integrating pH-responsive liposomes and an antifouling hydrogel. The hydrogel, synthesized from hydroxyethylacrylamide (HEAA) and diethylacrylamide (DEAA), mimics antibiofouling coating technology, forming a smooth hydrophobic surface that resists bacterial adhesion and maintains stable attachment in humid nasal environments [16]. Liposomes, composed of phospholipid bilayers and modified with pH-sensitive acylhydrazone bonds, encapsulate dexamethasone (DXMS). In acidic inflammatory conditions (pH~5.5), acylhydrazone cleavage triggers targeted drug release, while physiological pH preserves liposomal integrity to protect DXMS from enzymatic degradation [17,18,19,20]. DXMS, a glucocorticoid with low molecular weight and high lipophilicity, is ideal for liposomal encapsulation. Whereas short-term high-dose glucocorticoid administration may induce mucosal atrophy and increased tissue fragility, sustained low-dose delivery demonstrates therapeutic advantages by promoting epithelial regeneration and facilitating ordered extracellular matrix (ECM) deposition [21,22,23,24].

This system synergizes mechanical barrier properties, pH-triggered drug delivery, and antimicrobial activity. In vitro and in vivo studies demonstrate its ability to enhance mucosal repair, suppress inflammation, and regulate ECM dynamics through multiple signaling pathways. By combining antibiofouling strategies with stimuli-responsive drug release, HYD-Lip/DXMS@HG offers a potential solution for refractory nasal mucosal injuries. The synthesis and application of HYD-Lip/DXMS@HG are shown in Figure 1.

## 2. Materials and Methods

### 2.1. Materials

Hydroxyethylacrylamide (HEAA), diethylacrylamide (DEAA), lithium phenyl-2,4,6-trimethylbenzoylphosphinate (LAP) and N,N′-Methylenebisacrylamide (Bis) were purchased from Aladdin Chemistry Co, Ltd. (Shanghai, China). Polycarbonate membrane, dissolve dioleyl phosphatidylethanolamine (DOPE), carboxylated cholesterol and DSPE-HYD-PEG2000 were purchased from Xi’an Ruixi Biotechnology Co., Ltd. (Xi’an, China). DXMS was purchased from Shanghai Macklin Biochemical Technology Co., Ltd. (Shanghai, China). Phosphate buffer (0.1 mol/L, pH = 7.4), Modified Eagle’s medium (MEM), Cell Counting Kit-8 (CCK-8), and penicillin/streptomycin solution were from Servicebio Technology Co., Ltd. (Wuhan, China). Fetal bovine serum (FBS), Propidium Iodide (PI), Calcein Acetoxymethyl Ester (Calcein AM), and EDTA decalcified solution were purchased from Beyotime Biotechnology Co., Ltd. (Shanghai, China). Staphylococcus aureus solution was purchased from Shanghai Huzheng Biotechnology Co., Ltd. (Shanghai, China). Self-cross-linking hydrogel @SINUS was provided by Shanghai Sixth People’s Hospital (Shanghai, China).

### 2.2. Cell Cultures

The human nasal epithelial cell line (RPMI 2650) was purchased from Cyagen Biosciences Co., Ltd. (Santa Clara, CA, USA). This cell line was cultured at 37 °C with 5% CO_2_ in Modified Eagle’s medium (MEM) supplemented with 10% FBS and 1% penicillin–streptomycin mixture. The third to sixth generations of cells were used in the experiments, and the culture medium was changed every 2 to 3 d.

### 2.3. Preparation and Characterization of DXMS Liposomes

DXMS liposomes modified with acylhydrazone bond were synthesized with the thin-film hydration method followed by extrusion [25]. A total of 30 mg of DOPE, 10 mg of carboxylated cholesterol, and 10 mg of DSPE-HYD-PEG2000 were combined in 3 mL of chloroform. DXMS was dissolved in 500 μL of methanol via sonication in a water bath for 30 s. The chloroform and methanol solutions were then mixed and transferred to a 50 mL round-bottom flask. Organic solvents were removed using a rotary evaporator (R-210, Buchi, Flawil, Switzerland) under reduced pressure (≤10 mbar) at 40 °C, forming a uniform lipid film on the flask wall. Subsequently, 3 mL of deionized water was added to hydrate the lipid film. Processed by ultrasound and a lipid extruder (polycarbonate membrane, pore size 100 nm). The extruded liposomes were dialyzed using a nanodialysis device (polycarbonate membrane, pore size 10 nm) to remove unencapsulated DXMS. The final liposomal suspension was adjusted to a total volume of 5 mL with deionized water, yielding a DXMS concentration of 0.72 mg/mL. Add cryoprotectant and proceed with lyophilization. To characterize the morphology and size of liposomes, TEM was performed using a TF20 microscope (FEI, Hillsboro, OR, USA) operated at 200 kV acceleration voltage. Briefly, 5 μL of liposome suspension (1 mg/mL in deionized water) was deposited onto a carbon-coated copper grid (200 mesh) and allowed to adsorb for 2 min. Excess liquid was blotted with filter paper, followed by negative staining with 2 μL of 2% (*w*/*v*) phosphotungstic acid for 1 min to enhance contrast. The grid was air-dried and stored in a desiccator prior to imaging.

Size and zeta-potential were measured using nanoparticle size and zeta potential analyzer (NanoBrook 90plus PALS, Brookhaven Instruments Corporation, Holtsville, NY, USA). Liposome samples were diluted to 0.5 mg/mL in deionized water and equilibrated in a quartz cuvette for 5 min. Size distribution was reported as the mean ± standard deviation (SD) of three independent measurements, alongside the polydispersity index (PDI). Zeta-potential was measured by phase-analysis light scattering (PALS).

EE% was quantified using ultrafiltration centrifugation coupled with high-performance liquid chromatography (HPLC, Agilent 1260 Infinity). Free DXMS was separated from liposome-encapsulated DXMS using Amicon Ultra centrifugal filters (100 kDa MWCO) (Millipore Corporation, Burlington, VT, USA). The liposome suspension was centrifuged at 10,000× *g* for 30 min at 4 °C. The retentate (encapsulated fraction) was dissolved in methanol to disrupt liposomal membranes, and the supernatant (free drug fraction) was collected for analysis. HPLC analysis was performed on a C18 column (4.6 × 250 mm, 5 μM; Agilent Eclipse XDB-C18) with a mobile phase of methanol:water (70:30, *v*/*v*) at 1 mL/min flow rate and 240 nm detection wavelength. EE% was calculated as EE% = (W_total_ − W_free_)/W_total_ × 100%. W_total_ is the total drug added, and W_free_ is the free drug in the filtrate.

### 2.4. Preparation of Hydrogels

@HG was produced following the procedure as reported [16]. To summarize, 0.02 g of LAP and 0.02 g of Bis were dissolved in 1.6 mL of deionized water. Subsequently, 0.2 mL HEAA and 0.2 mL DEAA were added to the solution. The mixture was kept from light at 4 °C. HYD-Lip/DXMS@HG was fabricated by mixing liposomes and @HG by ultrasound and kept from light at 4 °C. The working concentration of the liposomes was 10 mg/mL, corresponding to a DXMS concentration of 0.72 mg/mL

### 2.5. Characteristics of the Hydrogels

The structures of @HG and HYD-Lip/DXMS@HG after UV irradiation were observed by SEM. The samples were previously lyophilized and subsequently placed on the objective tables prior to the application of gold sputtering. Subsequently, the sample morphology, energy spectrum mapping, and other tests were performed using the ZEISS GeminiSEM 300 scanning electron microscope with an accelerating voltage of 10 kV and a secondary electron detector.

Fourier transform infrared spectra (FTIR) were used to confirm the structure of the samples, over the range of 400–4000 cm^−1^ using the Fourier transform infrared spectrometer (Nicolet iS20, Thermo Fisher Scientific, USA).

Water contact angle test was performed to detect the hydrophobicity of hydrogels. A droplet of water is placed on the surface of the hydrogels, and the angle formed between the baseline of the droplet and the tangent at the droplet boundary is measured using the goniometer (JY-82C, Chengde Dingsheng, Chengde, China).

The rheological properties of hydrogels were characterized on the rheometer (HAAKE Mars60, Thermo Fisher Scientific, Karlsruhe, Germany) with a diameter of 10 mm plate and a thermostatic bath at the temperature of ~37.0 °C. Amplitude sweep tests were performed at a fixed angular frequency of 1 Hz to determine the linear viscoelastic region (LVR) and the variation in storage modulus (G′) and loss modulus (G″) with increasing shear stress.

The compression properties and the compression cyclic performance of the samples were measured using a universal testing machine (INSTRON 5982, Instron Corporation, Norwood, MA, USA). The samples were prepared as circular discs with a diameter of 5 mm and a thickness of 1 mm and tested at the temperature of ~37.0 °C. The compression tests were conducted at a crosshead speed of 0.5 mm/min, and the stress–strain curves were recorded. Each sample was subjected to compressive deformation until a strain of 80% was achieved or until failure occurred.

The adhesion performance of the hydrogels on different surfaces was evaluated using a rheometer (Haake Mars40, Thermo Scientific, Dreieich, Germany) in oscillatory mode. Oscillatory shear scanning was performed at an oscillatory frequency of 1 Hz with a fixed shear stress of 10 N, during which the storage (G′) and loss (G″) moduli as a function of oscillatory stress were recorded.

The swelling ratio of the hydrogels was determined by swelling tests. The hydrogel samples were immersed in PBS solution (pH = 7.4) at 37 °C and weighed from the solution without external water at various times (Wt). Hydrogels were weighed until swelling equilibrium was established. The swelling ratio was calculated via the following formula: Swelling ratio = (Mt − M0)/M0 × 100%. W0 and Wt represent the initial weight and the weight at different swelling times, respectively.

To analyze the release of DXMS from the HYD-Lip/DXMS@HG, HYD-Lip/DXMS and DXMS@HG, saline solutions with pH values of 5.5, 6.0, 6.5, and 7.5 were used to simulate the nasal microenvironment under normal (pH 6.0 ± 0.5) and inflammatory conditions (pH decreased to 5.5 or lower). The extreme acidic condition (pH 1.5) was included as a control to verify the acid-responsive mechanism. The hydrogels were immersed into 10 mL of saline solutions having pHs of 1.5, 5.5, 6.0, 6.5, and 7.5 under constant stirring (80 rpm, 37 °C). The HYD-Lip/DXMS was dispersed in 10 mL of saline solutions having pHs of 1.5, 5.5, 6.0, 6.5, and 7.5 under constant stirring (80 rpm, 37 °C). At a particular interval, the buffer samples were collected and concentrations of released DXMS were determined using a UV–Visible spectrophotometer at 240 nm. The same volume of the saline solution was added at the same pH as the amount of aliquot was taken out. The analysis was performed in triplicate. Cumulative release (%) was calculated as follows: Cumulative release (%) = (Mt/M0) × 100%. W0 and Wt represent the amount of DXMS actually loaded onto the HYD-Lip/DXMS@HG and the amount of released from the HYD-Lip/DXMS@HG at time t, respectively.

To characterize liposome integrity during release, cryo-transmission electron microscopy (Cryo-TEM) (FEI Tecnai Spirit TEM, Thermo Fisher Scientific, Hillsboro, OR, USA) was performed. HYD-Lip/DXMS@HG samples were immersed in PBS at pH 6.0 or 7.5 (37 °C, 80 rpm) for 24 h. Released fractions were collected, ultracentrifuged (100,000× *g*, 30 min), and resuspended in deionized water. A 3 μL aliquot was applied to a lacey carbon grid, blotted, and rapidly frozen in liquid ethane. Samples were observed using a FEI Talos F200C Cryo-TEM operating at 200 kV.

### 2.6. Biocompatibility of the Hydrogels

Prepare the hydrogels into discs with a diameter of 1 cm and a thickness of 0.1 cm, add 2 mL of MEM culture medium to each disc for extraction for 24 h to obtain the extract. Cultivate RPMI 2650 cells using the extraction solution.

The viabilities of hydrogels were evaluated by live/dead cell staining kit in vitro at the endpoint (3 days). The RPMI 2650 cells were washed by PBS three times, and then treated with propidium iodide (PI, 4 μM) and calceinAM (2 μM) in medium minus FBS at room temperature and then incubated in the darkness for 15 min. Afterwards, the cells were observed under fluorescent inverted microscope. The percentage of live and dead cells was calculated after counting by Image-J software (version 1.54g).

The cytotoxicity study was conducted using the CCK-8 assay. In a humidified incubator (37 °C, 5% CO_2_), RPMI 2650 cells were cultured in hydrogels’ extraction solution. The cells were digested with trypsin, diluted to a cell concentration of 1 × 10^5^ cells/mL and 100 μL of cell suspension was added to each well. Each group contained three replicate wells. The toxicity tests were performed on the first and second days. The extraction solution was replaced with fresh medium containing 10% CCK-8 solution at each time point and incubated for 2 h. The absorbance of each well was detected by an enzyme-labeled instrument.

### 2.7. Wound Healing Experiment

The monolayer cells were manually scratched by sterile 200 µL pipette tips. After being washed with PBS, cells were treated with specific extraction solution. Subsequently, digital photos of the wound area were captured by optical microscope at 0 h, 24 h, and 48 h, respectively. The cell-free area was measured and analyzed by Image-J.

### 2.8. In Vitro Antibacterial Activity

In vitro antibacterial activity of the hydrogels was assessed using S. aureus. Bacterial suspensions in the exponential growth phase were diluted to 1 × 10^6^ CFU/mL and centrifuged. A sterile PBS solution was added to resuspension solution. The hydrogels were soaked with a diameter of 10 mm and a thickness of 1 mm in PBS until absorption balance was reached, then disinfected with ultraviolet light for 1 h, and then immersed in bacterial suspension. After 4 h of incubation at 37 °C, 100 μL bacterial solution was aspirated, diluted, seeded on agar plates, and cultured overnight at 37 °C. Finally, the bacterial colonies were observed and counted.

### 2.9. In Vivo Nasal Mucosal Injury Surgical Procedure

The animal study protocol was approved by the Ethics Committee of Shanghai Sixth People’s Hospital Affiliated to Shanghai Jiao Tong University School of Medicine (No: DWLL2024-1052). Mature New Zealand white rabbits (6 per group, *n* = 24 total; male, 2.5 ± 0.5 kg) were split into 4 groups at random: CTRL, @SINUS, @HG, and HYD-Lip/DXMS@HG.

On the nasal dorsum, an area of 2 × 3 cm was shaved and cleaned with povidone-iodine. Then, a vertical midline incision was made along the back of the nasal dorsum to expose the nasal bone and drill a hole with a diameter of about 6 mm using a bone drill. Upon exposing the nasal septum, the mucosa was incised and peeled, revealing a diameter of approximately 6 mm. The injured site was then treated with hydrogels. The skin incision was closed with a non-absorbable suture. Intramuscular injection of penicillin was administered for 3 consecutive days after the operation.

### 2.10. Histology, Immunohistochemistry, and Immunofluorescence Evaluation

The tissues were fixed in 4% paraformaldehyde, decalcified, dehydrated, paraffin-embedded, and sectioned into 4 μM thickness slices. H&E and Masson’s trichrome staining were performed, and immunohistochemistry staining was carried out to detect the expression of TNF-α (1:1000, Servicebio). Iimmunofluorescence staining was carried out to detect the expression of CD31 (1:1000, Servicebio). Tissue slices were scanned using an Aperio AT2 scanning system (Leica, Berlin, Germany), and quantitative analysis was performed using Image-J software.

### 2.11. RNA Sequencing

We collected the new and peripheral tissues of nasal septum wounds in control, @HG Sinus, @HG, HYD-Lip/DXMS@HG groups after 7 days post-surgery. Total RNA was extracted with TRIzol reagent (Beyotime, China). Pathway enrichment was performed with KEGG pathway analysis (OE Biotech, Shanghai, China).

### 2.12. Statistical Analysis

The data were derived from at least three independent experiments and all obtained data were expressed as mean value ± standard deviation (SD). One-way variance analysis (ANOVA) followed by Tukey’s post hoc test was used for multiple comparisons. Single, double, triple, and four asterisks represent *p* ˂ 0.05, 0.01, 0.001, and 0.0001, respectively. Statistical analyses were performed using GraphPad Prism 5 and Origin 2021.

## 3. Results

### 3.1. Characterization of Liposomes

DXMS exhibits poor water solubility, and short-term high-dose administration can cause mucosal damage, whereas low-dose continuous administration promotes mucosal repair [26,27]. The phospholipid bilayer of liposomes provides an ideal compartment for storing lipophilic drugs, making liposomes an effective carrier for DXMS to enhance its solubility [28]. The nasal cavity is a weakly acidic environment with an average pH of 6.0 ± 0.5, which further decreases under inflammatory conditions [29,30]. Leveraging the weakly acidic environment of the nasal cavity, we designed acylhydrazone bond-modified liposomes. These bonds cleave in a weakly acidic environment, causing the liposomes to rupture and release the encapsulated drugs.

The TEM morphology of the liposomes is shown in Figure 2a. The liposomes were spherical with honeycomb-like microcavities. The encapsulation efficiency of DXMS liposomes is approximately 90%, with drug loading rates of 7.16%, 7.08%, and 7.36%, respectively. The hydrodynamic size ranges from 149.52 nm to 163.78 nm. The Zeta potential was within the range from −15 mV to −20 mV, which is considered a favorable potential for assuring particle stability (Table 1, Figure 2b). HPLC results show that, after separating free DXMS from the liposomes by ultrafiltration centrifugation and eliminating the interference of lipid components via chromatographic separation, a 240 nm absorption peak consistent with DXMS standard was observed, confirming successful DXMS encapsulation (Figure 2c) [31,32].

### 3.2. Characterization of Hydrogels

The hydrogel was initially developed as an anti-biofouling coating, featuring excellent properties for preventing blood cell adhesion [16]. Its main components are hydroxyethylacrylamide (HEAA) and diethylacrylamide (DEAA). We tested the cell adhesion properties of hydrogels synthesized with different ratios of HEAA and DEAA and found that the hydrogel synthesized at a 1:1 ratio exhibited the strongest anti-blood cell adhesion capability (Appendix A).

Scanning electron microscopy (SEM) images (Figure 3a) revealed the microgram of hydrogel and integration of hydrogel and liposomes. The polyacrylamide-based copolymer hydrogel featured a smooth surface and minuscule intervals. These attributes enable it to repel bacteria and impurities. Liposomes were evenly dispersed in the hydrogel.

In Figure 3b, the FTIR spectrum of the HYD-Lip/DXMS contained a broad absorption band around 3350 cm^−1^, which depicts the stretching vibrations of the –N–H– group. The peak at 1740 cm^−1^ for both HYD-Lip/DXMS and HYD-Lip/DXMS@HG corresponds to the vibrational frequency of the C=O bond, indicative of the ester groups within the liposomes. The presence of the band at 1630 cm^−1^ of HYD-Lip/DXMS and HYD-Lip/DXMS@HG was attributed to the stretching vibration of –C=N–, a characteristic feature of the acylhydrazone bond. After the incorporation of liposomes, the FTIR spectrum of the hydrogel did not change significantly. The water contact angles at the @HG and HYD-Lip/DXMS@HG surfaces were 105.07° and 103.57°, respectively, indicating the hydrophobic properties (Appendix A).

Rheological analysis demonstrated time-dependent mechanical evolution in photocross-linked hydrogels. The storage modulus of soft tissues is typically in the range of 1–100 kPa [33,34]. In this study, the storage modulus (G′) of the hydrogels was determined to be 9000–18,000 Pa (Figure 4a,b), which falls well within the range compatible with soft tissues. This characteristic enables the hydrogel to effectively adapt to the dynamic deformations of mucosal tissues—such as the periodic movements occurring during respiration—while minimizing potential adverse effects on surrounding tissues.

Figure 5a compares the compressive mechanical profiles of @HG and HYD-Lip/DXMS@HG under 30 s, 1 min and 5 min photo-cross-linking. The 1 min cross-linked hydrogels exhibited significantly higher stress values than their 30 s counterparts; however, the stress values of the 5 min cross-linked gels decreased significantly. Incorporation of liposomes reduced the overall mechanical strength, with HYD-Lip/DXMS@HG showing lower stress magnitudes compared to @HG at equivalent cross-linking durations. Despite this strength reduction, the composite hydrogels demonstrated enhanced ductility at high-strain regimes (>60% strain). Specifically, the 30 s cross-linked HYD-Lip/DXMS@HG (green curve) exhibited a more gradual stress escalation and delayed fracture initiation, indicative of enhanced energy dissipation mechanisms through dynamic network reconfiguration—a hallmark of resilience optimization in viscoelastic biomaterials. As evidenced by the cyclic compression tests in Figure 5b, 1 min photo-cross-linking duration markedly enhanced hydrogel mechanical robustness. While liposomes incorporation compromised ultimate compressive strength, the composite hydrogel exhibited superior elastic recovery. Notably, beyond 20% strain, @HG displayed rapid stress escalation, while the stress growth of HYD-Lip/DXMS@HG was relatively gentle, showing certain resilience optimization.

We evaluated the adhesiveness of hydrogels to mucosa and cartilage using rheological measurements. The G′ of the hydrogels on both mucosa and cartilage remained relatively stable over the measured time period, suggesting stable adhesive properties of hydrogels, with stronger adhesion observed on the cartilage compared to the mucosa (Figure 6). This suggests that, in cases of severe mucosal damage with exposed cartilage, hydrogels adhered more firmly at the injured site. Furthermore, HYD-Lip/DXMS@HG demonstrated better adhesion performance than @HG.

The DXMS was released in two stages: the release of liposomes from hydrogel and the release of DXMS from liposomes. The release of liposomes from hydrogel was related to the swelling or degradation of hydrogel, with some liposomes flowing out through the pores of the hydrogel, while another part was released as the hydrogel swelled and degraded. Cryo-TEM analysis provided direct morphological evidence for the two-stage release mechanism. At pH 7.5, particles released from HYD-Lip/DXMS@HG matched the original liposome size and spherical structure, confirming that liposomes were released intact from the hydrogel without premature drug release (stage 1) (Figure 7a). In contrast, at pH 6.0, fragmented liposomal membranes were observed, indicating acid-triggered rupture of acylhydrazone-modified liposomes, which released encapsulated DXMS (stage 2) (Figure 7b). These findings align with the pH-responsive release profile (Figure 8a), where minimal DXMS release occurred at pH 7.5 (intact liposomes retained drug) and accelerated release at pH 6.0. 

As the liposomes developed in this study demonstrated pH-sensitive characteristics, various pH solutions were employed to assess the drug release profile. The cumulative release profiles of DXMS from HYD-Lip/DXMS@HG at different pH (1.5, 5.5, 6.0, 6.5, 7.5) conditions are shown in Figure 8a. The results indicate that pH plays a key role in the release profile and, at pH 1.5, a highest amount of DXMS was released from HYD-Lip/DXMS@HG. Within solutions characterized by pHs of 5.5, 6.0, and 6.5, which corresponds to the mildly acidic conditions of the nasal cavity, the DXMS’s release is notably decelerated. Conversely, in a neutral environment with a pH of 7.5, the DXMS’s release is minimal, as anticipated. The cumulative release profiles of DXMS from liposomal dispersion (HYD-Lip/DXMS) and free DXMS-loaded hydrogel (DXMS@HG) at different pH (1.5, 5.5, 6.0, 6.5, 7.5) conditions are shown in Appendix A. For the HYD-Lip/DXMS, the absence of hydrogel encapsulation led to rapid rupture and drug release under acidic conditions due to acylhydrazone bond cleavage (Appendix A). In the case of DXMS@HG, the low molecular weight of DXMS facilitated its easy diffusion from the gel, and it lost pH responsiveness (Appendix A). The composite HYD-Lip/DXMS@HG showed the best sustained-release performance in comparative analysis. These results collectively demonstrate the effectiveness of the encapsulation strategy.

Swelling plays a central role in solute diffusion coefficient, surface properties, adhesion, and mechanical integrity. The hydrophilic polyethylene glycol (PEG) chain in MPEG2K-HYD-DSPE liposomes facilitates the incorporation of additional hydrophilic moieties into the hydrogel matrix, consequently enhancing the hydrogel’s swelling capacity through improved hydration effects [35]. The swelling behaviors of hydrogels is shown in Figure 8b. After the introduction of liposomes, the swelling ratio of hydrogels increased from 42.48% ± 0.62% to 44.33% ± 0.17%.

### 3.3. Hydrogels in Cellular Functions and Their Inhibition of Bacterial Production

The cellular functions of hydrogels were analyzed utilizing the human nasal cell line RPMI 2650 to simulate the actual nasal mucosa [36,37]. Live/dead staining was acquired to assess the cytocompatibility of hydrogels. The percentage of living cells of the @HG and HYD-Lip/DXMS@HG groups was slightly higher than that of the control group, indicating that the hydrogels exhibited good cytocompatibility (Figure 9a). After 48 h of coculture with hydrogels, cells in the @HG and HYD-Lip/DXMS@HG groups behaved better than the control group in the cell scratch experiment (Figure 9b,d).

The toxicity of polymeric materials presents a significant drawback in the development of drug delivery vehicles. For a matrix to be considered a safe drug carrier, it must exhibit low toxicity. The cytotoxicity of hydrogels was tested by CCK8 assay. The cell was cultured with extraction solution of hydrogels. After 24 h, 48 h, and 72 h, the cell proliferation rate was determined. The results reveal that @HG and HYD-Lip/DXMS@HG possesses an excellent cell proliferation rate (Figure 9c). The results of the antibacterial tests reveal the varying impacts of different materials on the proliferation of Staphylococcus aureus (Figure 9e,f). The DEAA and HEAA groups displayed minimal antibacterial efficacy. Nevertheless, upon combining DEAA and HEAA to synthesize hydrogels, a substantial augmentation in antibacterial capacity was observed. The HYD-Lip/DXMS@HG group, while slightly less effective, still showcased commendable antibacterial characteristics. These results indicate that the prepared hydrogels exhibit significant biocompatibility and antibacterial properties, and thus can be regarded as a non-toxic controlled drug delivery system.

### 3.4. Hydrogels Accelerated Nasal Wound Repair In Vivo

We performed nasal septum mucosal injury surgery on rabbits. The nasal septum mucosa was taken for RNA sequencing on day 7 after surgery, and histological examination was performed on day 14 after surgery (Figure 10a). The general procedure of the operation is shown in Appendix A. Rabbits were divided into four groups and treated with saline solution, @SINUS, @HG, or HYD-Lip/DXMS@HG, respectively. @SINUS, a self-cross-linked hydrogel predominantly composed of sodium hyaluronate, is commonly utilized as a clinical packing material following sinus surgery [38].

On day 7 (Appendix A) and day 14 (Figure 10b), distinct groups of wounds were observed. The group receiving treatment with @HG or HYD-Lip/DXMS@HG exhibited significantly superior outcomes compared to the control group. The group treated with @SINUS also existed better outcomes than the control group. H&E staining revealed intact mucosa with no significant edema, along with evidence of regenerating cilia in @HG and HYD-Lip/DXMS@HG groups (Figure 10c,d). The findings suggest that both @HG and HYD-Lip/DXMS@HG facilitated the healing process of nasal mucosa injuries.

Masson’s trichrome staining revealed that the mucosa of the control group appeared disrupted, with a disordered collagen deposition and structural disorganization. The staining in the @SINUS group showed a better-organized structure with loose mucosa and decreased collagen deposition beneath the epithelial layer. The mucosa in the @HG group exhibited improved structural integrity compared to the @SINUS group, with collagen beginning to align directionally, indicating ongoing extracellular matrix (ECM) remodeling. Additionally, partial restoration of normal tissue architecture was observed. The mucosa in the HYD-Lip/DXMS@HG group exhibited the most notable improvement, with continuous epithelial integrity, enhanced collagen deposition, and near-complete restoration of normal tissue structure (Figure 10c,e). Immunohistochemical staining was performed to detect TNF-α, allowing for the evaluation of inflammation levels across four distinct groups. The HYD-Lip/DXMS@HG group exhibited a significantly stronger anti-inflammatory effect compared to the other groups (Figure 10c,f).

Immunofluorescence staining of CD31 and α-SMA demonstrated a significant increase in vascular density within the regenerated mucosa, notably highlighting the enhanced wound healing performance in both @HG and HYD-Lip/DXMS@HG groups (Figure 11a,b,f,g). CD45 immunofluorescence analysis revealed reduced leukocyte infiltration in the regenerated mucosa of @HG and HYD-Lip/DXMS@HG groups compared to control and @SINUS groups (Figure 11c,h), suggesting effective suppression of inflammatory responses at the injury site. PCNA immunofluorescence quantification showed greater relative fluorescence intensity in @HG and HYD-Lip/DDXMS@HG groups versus controls, indicating elevated cellular proliferation activity in nasal mucosa (Figure 11d,i). This enhanced proliferative capacity implies that the hydrogel potentially provides a favorable microenvironment for damaged tissue regeneration. Importantly, upregulated Tubulin expression observed in @HG and HYD-Lip/DXMS@HG groups indicates enhanced cytoskeletal dynamics and improved cell migratory capacity, which may collectively contribute to tissue regeneration and repair processes (Figure 11e,j).

### 3.5. Sequencing Results of Mucosal Repair

To investigate the molecular mechanisms underlying hydrogel-mediated nasal mucosa regeneration, we performed RNA sequencing on mucosal tissues collected from four experimental groups at day 7 post-treatment. Principal component analysis (PCA) revealed distinct transcriptomic profiles among HYD-Lip/DXMS@HG, @HG, @SINUS, and control groups (Figure 12a), suggesting potential modulation of regenerative signaling pathways by both the hydrogel matrix and encapsulated HYD-Lip/DXMS. The volcano plot displays substantial differences with 1973 differentially expressed genes (*p* < 0.05) and 507 significantly different genes between HYD-Lip/DXMS@HG group and control group (Figure 12b). The top 20 significantly altered genes were functionally enriched in ECM–receptor interaction, TGF-*β* signaling, Hippo signaling, and PI3K-Akt pathways (Figure 12c,d). Subsequent targeted analysis focused on mRNAs associated with ECM remodeling, immune homeostasis, and Epitheliosis (Figure 12e,f).

Among upregulated genes, ECM regulators *THBS1* and *FBLN5* demonstrated collagen organization functions—*THBS1* through TGF-*β* activation and *FBLN5* via *MMP* inhibition—consistent with improved collagen alignment observed in Masson staining [39,40,41]. Notably, the significant upregulation of *MMP7* suggests a dual regulatory mechanism in ECM homeostasis: while promoting epithelial migration through laminin cleavage, *MMP7* simultaneously activates antimicrobial defenses via prodefensin processing [42,43]. This functional coupling may to some extent explain the material’s ability to facilitate structural repair without secondary infection—a common failure mode of traditional fillers like gelatin sponges due to biofilm formation. Comparative analysis with @SINUS group revealed distinct anti-inflammatory properties: while @SINUS upregulated pro-inflammatory pathways (TNF, NF-κB, JAK-STAT), both @HG and HYD-Lip/DXMS@HG groups exhibited significant suppression of these pathways (Appendix A). Mechanistically, *YWHAG* (14-3-3 family member) upregulation correlates with increased mucosal thickness in H&E staining, potentially through cytoplasmic sequestration of *YAP* to balance epithelial proliferation/differentiation [44,45]. Concurrent activation of PI3K-Akt (via *IRS2*) and Wnt/*β*-catenin (via *LRP6*) pathways suggest synergistic promotion of cell survival and epithelial migration [46,47,48].

Key downregulated genes included *TLR9*, whose suppression correlates with NF-κB inactivation and reduced pro-inflammatory cytokine release [49]. The inhibition of *COL6A6* aberrant expression indicates antifibrotic effects through ECM—receptor interaction modulation. These findings collectively demonstrate that HYD-Lip/DXMS@HG achieves synchronized structural—functional mucosal restoration via multi-pathway coordination: dynamic ECM remodeling, inflammation resolution, epithelial regeneration, and angiogenesis.

## 4. Discussion

This study developed a pH-responsive liposome–hydrogel composite (HYD-Lip/DXMS@HG) that enables acid-triggered DXMS release via acylhydrazone bond-modified liposomes, combined with the mechanical stability and antifouling properties of a dual-network hydrogel (HEAA/DEAA). In vitro experiments demonstrated sustained drug release, enhanced RPMI 2650 cell migration, and antibacterial efficacy. In vivo, the composite suppressed inflammation by downregulating TNF-α and CD45+ leukocyte infiltration, while activating ECM receptor interaction, Hippo, and PI3K-Akt pathways to promote collagen remodeling and angiogenesis.

Compared to conventional hydrogels (e.g., hyaluronate-based @SINUS) with passive drug release and static mechanical support, our composite achieves dynamic drug delivery via pH-responsive liposomes, addressing microenvironment-specific regulation challenges. The HEAA/DEAA dual-network design outperforms traditional antifouling materials by combining low cell adhesion with enhanced mucosal adhesion, overcoming the detachment issue of existing materials on wet tissues [50,51]. Furthermore, RNA-seq revealed *MMP7*/TGF-*β* crosstalk, enabling coupled ECM remodeling and antimicrobial functions—a significant advance over single-pathway targeting approaches.

This work pioneers the integration of pH-responsive liposomes with antifouling hydrogels, establishing a “dynamic drug release-mechanical support-antimicrobial barrier” tripartite synergy that redefines mucosal repair material design (e.g., expandable to other inflammation-targeted drugs via acylhydrazone bonds). Clinically, the composite’s rapid epithelial regeneration and antifibrotic effects in rabbits address refractory conditions like septal perforation and post-sinusitis repair. Theoretically, *YWHAG*-mediated cytoplasmic sequestration of *YAP* illuminates spatiotemporal regulation of the Hippo pathway in mucosal regeneration, enriching mechanistic understanding [52].

The limitations of this study primarily reside in the translational relevance between experimental models and clinical complexity. Firstly, anatomical disparities between rabbit and human nasal cavities may obscure precise interpretation of material–mucosa interface interactions [53,54]. Secondly, the current antimicrobial evaluation—limited to single-exposure acute infection models—does not capture the recurrent microbial challenges characteristic of chronic rhinosinusitis, nor does it address how interindividual heterogeneity in inflammatory microenvironments (e.g., pH fluctuations, protease concentration variations) regulates drug release kinetics [55]. To overcome these barriers, future work could employ miniature pig nasal models that emulate human mucus secretion patterns and airflow dynamics, coupled with the development of microenvironment-adaptive hydrogels incorporating reactive oxygen species-responsive elements for dynamic mechanical modulation [56,57]. Concurrent establishment of spatial transcriptomic–metabolomic integration platforms would enable systematic decoding of material-induced immune microenvironment remodeling, ultimately advancing personalized mucosal regeneration strategies.

## 5. Conclusions

This study developed an innovative composite system (HYD-Lip/DXMS@HG) integrating pH-responsive liposomes with anti-biofouling hydrogel, effectively addressing the clinical challenge of refractory nasal mucosal repair. The polyacrylamide-based copolymer hydrogel demonstrates superior mucosal adhesion and hydrophobic surface characteristics, enabling both ciliary clearance resistance and bacterial colonization inhibition in humid nasal environments. The acylhydrazone-modified liposomes achieve pH-responsive DXMS release, with sustained low-dose delivery significantly promoting epithelial regeneration and collagen organization while preventing mucosal atrophy. Animal studies confirm HYD-Lip/DXMS@HG’s superior therapeutic outcomes, with treated groups showing enhanced mucosal restoration, antimicrobial efficacy, and collagen alignment compared to controls. Transcriptomic profiling reveals multi-modal regenerative mechanisms: *THBS1*/*FBLN5* upregulation drives ECM-TGF-*β* axis-mediated matrix remodeling, *TLR9*/*IRF4* suppression resolves inflammatory signaling, while coordinated *YWHAG*-Hippo/*YAP* regulation and *LRP6*-Wnt activation balance cellular proliferation/differentiation. This bioengineered strategy establishes a promising alternative to conventional nasal packing and drug delivery systems, offering synchronized structural and functional mucosal healing.

## Figures and Tables

**Figure 1 pharmaceutics-17-00690-f001:**
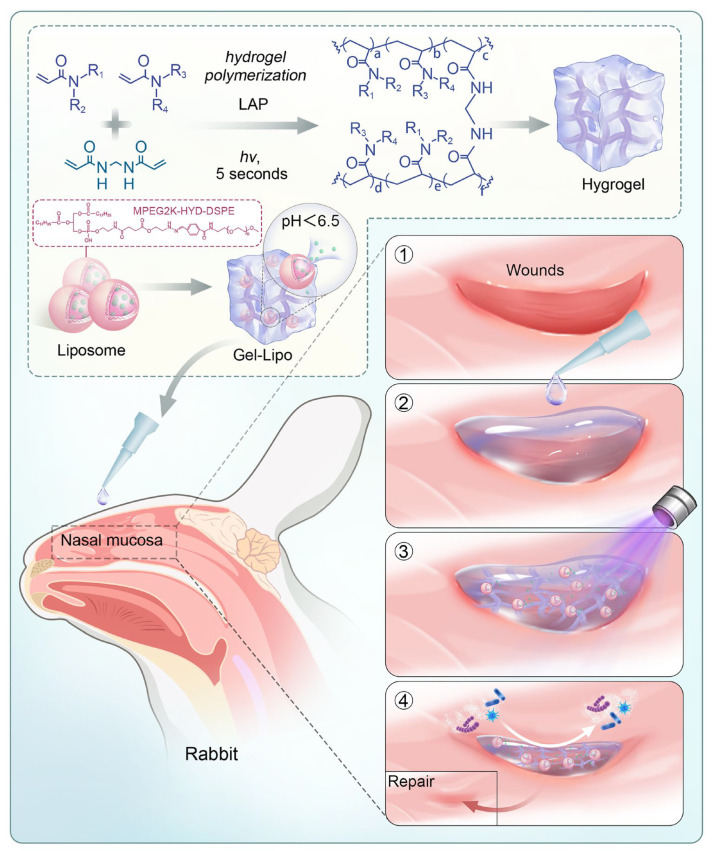
Schematic diagram of the preparation of HYD-Lip/DXMS@HG and application in rabbits. Abbreviations: LAP: lithium phenyl-2,4,6-trimethylbenzoylphosphinate; MPEG2K: Methoxy polyethylene glycol 2000; HYD: Hydrazone; DSPE: Distearoylphosphatidylethanolamine.

**Figure 2 pharmaceutics-17-00690-f002:**
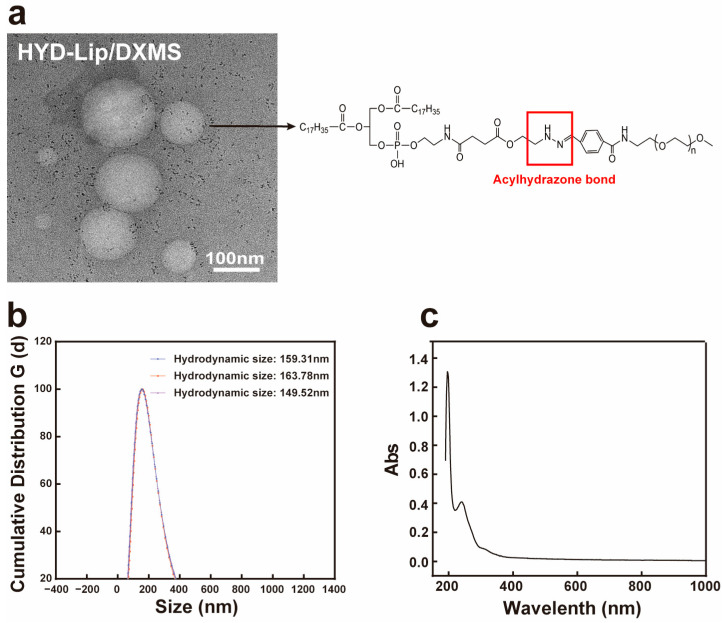
Characteristics of liposomes. (**a**) Transmission electron micrograph of acylhydrazone bond modified DXMS liposome. Scale bar: 100 nm; (**b**) The hydrodynamic size of DXMS liposome; (**c**) The UV spectrum of DXMS standard.

**Figure 3 pharmaceutics-17-00690-f003:**
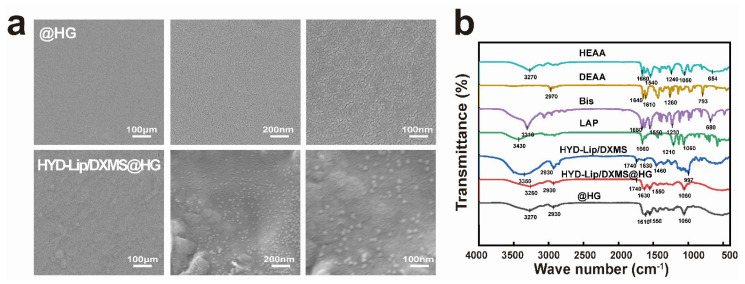
Characteristics of hydrogels. (**a**) Scanning electron micrograph of @HG and HYD-Lip/DXMS@HG. Scale bar: 100 nm; (**b**) FTIR spectrum of HEAA, DEAA, Bis, LAP, HYD-Lip/DXMS, HYD-Lip/DXMS@HG, @HG.

**Figure 4 pharmaceutics-17-00690-f004:**
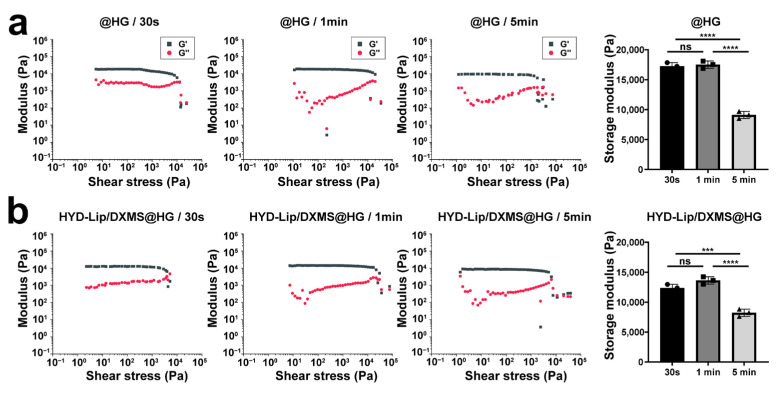
Gelation kinetics of @HG and HYD-Lip/DXMS@HG with varied cross-linking time. (**a**) Storage moduli of @HG with varied cross-linking time (e.g., 30 s, 1 min, 5 min) where G′ and G″ represent storage and loss moduli, respectively; (**b**) Storage moduli of HYD-Lip/DXMS@HG with varied cross-linking time. Data are expressed as mean ± SD (*n* = 3). One-way variance analysis followed by Tukey’s post hoc test was used to indicate the significance, and *** *p* < 0.001, **** *p* < 0.0001, not significant (ns).

**Figure 5 pharmaceutics-17-00690-f005:**
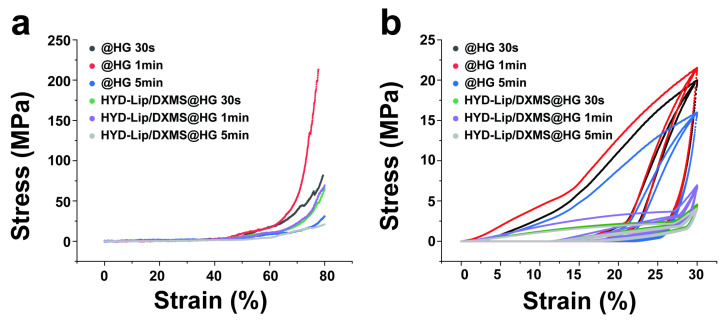
Stress–strain curves of @HG and HYD-Lip/DXMS@HG. (**a**) Compressive stress–strain curves of @HG and HYD-Lip/DXMS@HG with varied cross-linking times; (**b**) Cyclic compression behavior (3 cycles) of @HG and HYD-Lip/DXMS@HG under varied cross-linking times.

**Figure 6 pharmaceutics-17-00690-f006:**
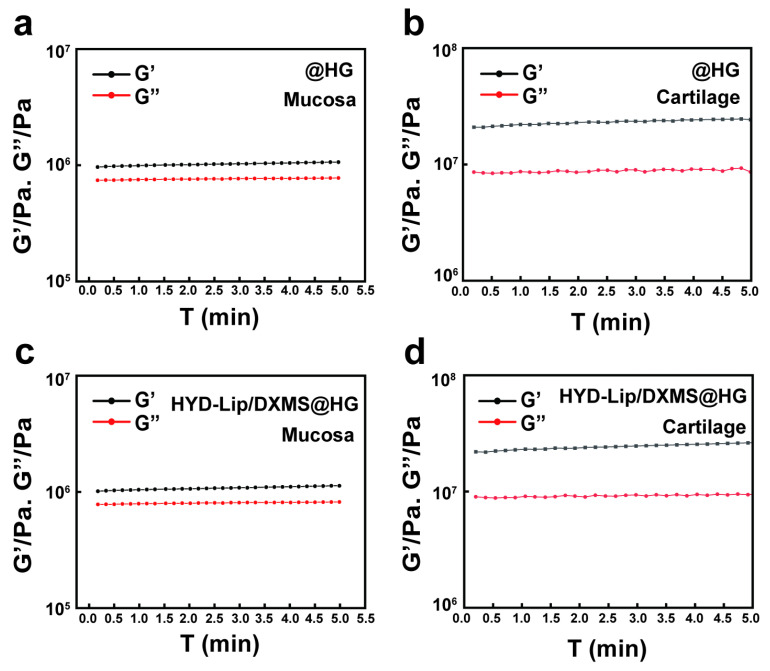
Adhesion test of @HG and HYD-Lip/DXMS@HG. (**a**,**b**) Rheological behavior of @HG adhered to mucosa and cartilage; (**c**,**d**) Rheological behavior of HYD-Lip/DXMS@HG adhered to mucosa and cartilage.

**Figure 7 pharmaceutics-17-00690-f007:**
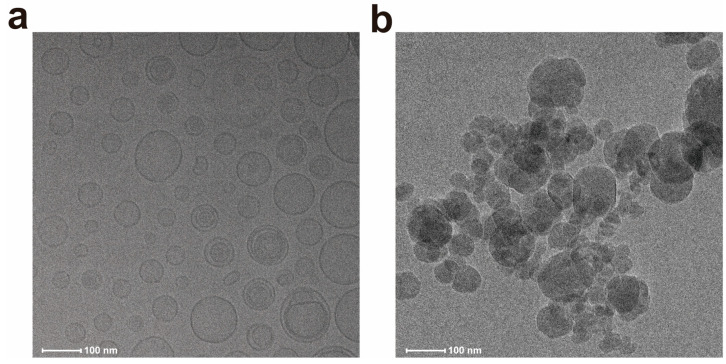
Cryo-TEM images of liposomes released from hydrogel. (**a**) Intact liposomes released at pH 7.5; (**b**) Ruptured liposomes released at pH 6.0.

**Figure 8 pharmaceutics-17-00690-f008:**
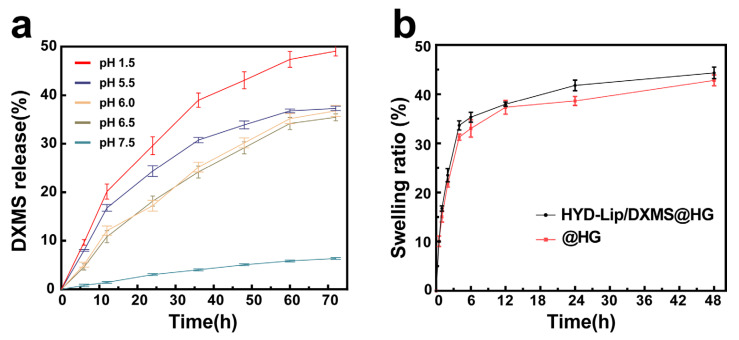
(**a**) DXMS release of HYD-Lip/DXMS@HG in pH 1.5, pH 5.5, pH 6.0, pH 6.5, pH 7.5; (**b**) Swelling ratio of @HG and HYD-Lip/DXMS@HG.

**Figure 9 pharmaceutics-17-00690-f009:**
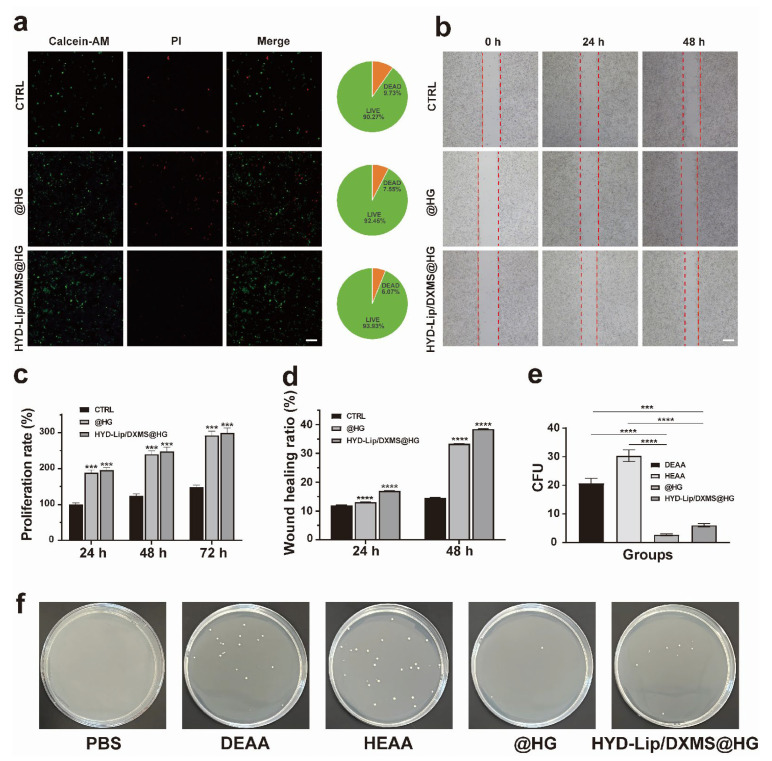
In vitro biocompatibility evaluation and antibacterial testing of hydrogels. (**a**) Images and quantitative analysis of live/dead fluorescence staining. Scale bar: 50 μM; (**b**) Effect of HYD-Lip/DXMS@HG on RPMI 2650 evaluated by the wound healing experiments. The red dot lines are used to mark the scratch area for facilitating the observation of cell migration at 0 h, 24 h, and 48 h under different conditions (CTRL, @HG, HYD-Lip/DXMS@HG). Scale bar: 100 μM; (**c**) Statistical analysis of proliferation rate of RPMI 2650 after different treatments; (**d**) Statistical analysis of RPMI 2650 mobility after different treatments; (**e**,**f**) Antibacterial properties of materials. *** *p* < 0.001, **** *p* < 0.0001.

**Figure 10 pharmaceutics-17-00690-f010:**
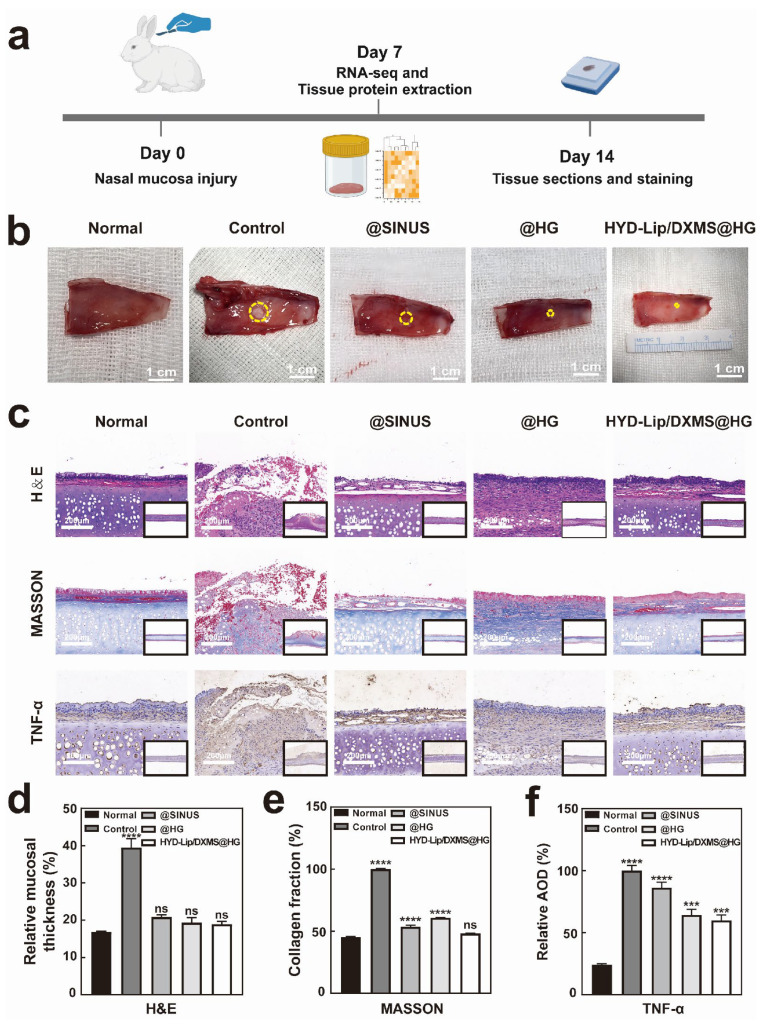
Results of rabbit nasal mucosa after repair. (**a**) Schematic diagram of animal experiments; (**b**) Repaired mucosae obtained from different treatment groups on day 7. The yellow circles highlight the size of the nasal mucosal wounds. Scale bar: 1 cm; (**c**) Hematoxylin and eosin (H&E) staining, Masson staining and TNF-α immunohistochemistry staining at the wound site on day 14 after different treatments. Scale bar: 200 μM; (**d**) On day 14, the ratio of mucosal thickness on the injured side to the full thickness of the nasal septum; (**e**) Relative content of collagen in nasal mucosa by Masson staining on day 14; (**f**) Quantitative data of relative average optical density (AOD) of TNF-α. *** *p* < 0.001, **** *p* < 0.0001, not significant (ns).

**Figure 11 pharmaceutics-17-00690-f011:**
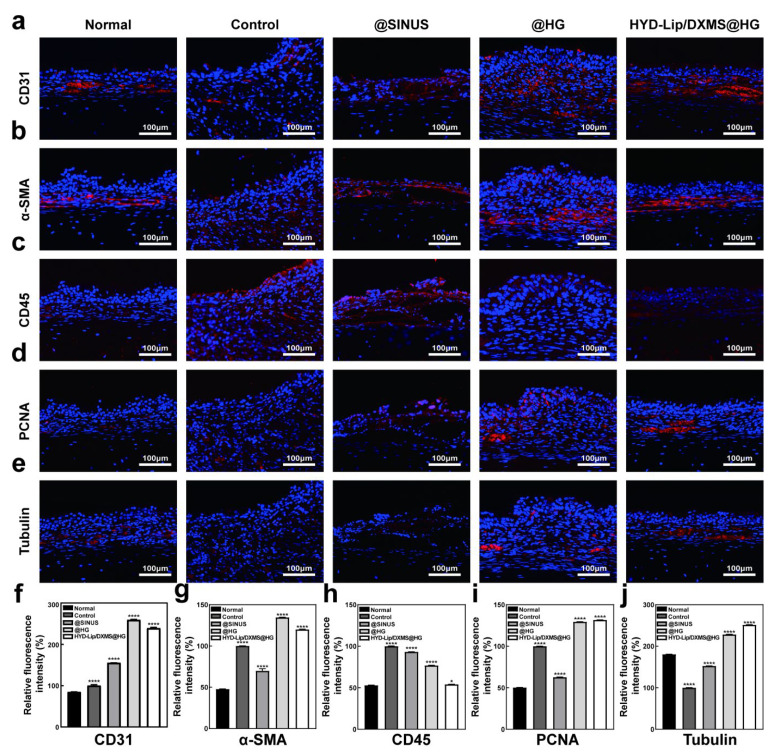
Results of immunofluorescence staining. (**a**–**e**) CD31, α-SMA, CD45, PCNA, and Tubulin immunofluorescence staining pictures taken on day 14 after different treatments. Scale bar: 100 μM; (**f**–**j**) Quantitative data of relative fluorescence intensity of CD31, α-SMA, CD45, PCNA and Tubulin. * *p* < 0.05, **** *p* < 0.0001.

**Figure 12 pharmaceutics-17-00690-f012:**
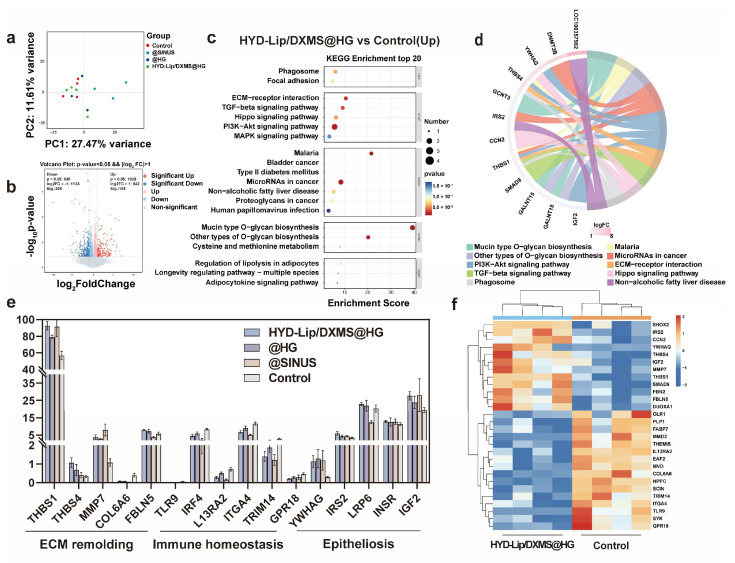
Analysis of RNA sequencing results of nasal injury and repair sites in rabbits at early stage (7 days). (**a**) Principal component analysis (PCA) of RNA sequencing results at 7 days for the control group and other experimental groups; (**b**) Volcano plot depicting differentially expressed genes between the HYD-Lip/DXMS@HG group and the control group at 7 days; (**c**,**d**) Top 20 upregulated signaling pathways in the KEGG enrichment analysis for the HYD-Lip/DXMS@HG group and the control group at 7 days; (**e**) Gene expression levels related to ECM remolding, Immune homeostasis and Epitheliosis at 7 days; (**f**) Heatmap of gene expression related to ECM remolding, Immune homeostasis and Epitheliosis in the HYD-Lip/DXMS@HG group and the control group at 7 days. Data are shown as mean ± SEM (*n* = 3).

**Table 1 pharmaceutics-17-00690-t001:** Data of DXMS liposomes.

DXMS Liposome	Encapsulation Rate (%)	Loading Efficiency (%)	Hydrodynamic Size (nm)	PDI	Zeta Potential (mV)
1	90.27	7.16	159.31	0.223	−18.42
2	90.49	7.08	163.78	0.216	−17.70
3	89.75	7.36	149.52	0.215	−19.04

## Data Availability

The data presented in this study are within the article and Appendix A or upon request from the corresponding authors.

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
