# Peer review of "pH-Responsive Liposome–Hydrogel Composite Accelerates Nasal Mucosa Wound Healing"

_pharmaceutics, 2025, doi:10.3390/pharmaceutics17060690_

Round 1
Reviewer 1 Report
Comments and Suggestions for Authors
This study developed a pH-responsive liposome-hydrogel composite (HYD-484 Lip/DXMS@HG which can release dexamethasone (DXMS) release by acid-triggered environment in nasal. The author did in vitro and in vivo characterization of their developed hydrogel products and showed wound healing effect after applied HYD-484 Lip/DXMS@HG in rabbit model. The author also did RNA sequencing study to understand the molecular mechanisms of the nasal tissue healing process. However, the current reviewer suggests there are some questions need to be addressed before the final approval.
- Please explain why conducting rheological study for @HG and HYD-Lip/DXMS@HG at 25°C? The reviewer recommends using relevant physiological temperature (e.g., 37°C) for this study, as well as the compression property characterization of the @HG and HYD-Lip/DXMS@HG.
- The author states the nasal cavity has average pH of 6.0±0.5. However, the in vitro release study using buffer at pH 1.5, pH 5.5, and pH7.5. Provide justification why the release study did not conduct at pH 6.0±0.5.
- For compressive mechanical study (figure 5), why there is no study for 5 minutes time point after gelation?
- The author characterized the mechanical property of the gel product. Please add a justification to explain the specific mechanical property won’t affect the surrounded tissue after applied the product to the target disease area.
- The author states the release of DEX has two stages: (1) the release of liposomes from hydrogel and (2) the release of DEX from liposomes. However, there are lack of evidence of the two-stage release mechanism.
- The author does not show the liposome stability in the hydrogel. Please provide justification that the liposome can keep integrity after loaded into the hydrogel that the DEX won’t leak out before the liposome released out.
- Provide a characterization to demonstrate the released particle from the HYD-Lip/DXMS@HG is liposome (e.g., cryo-TEM).
Reviewer 2 Report
Comments and Suggestions for Authors
Yang et al.. are reporting a pH-responsive liposome-hydrogel composite loaded with dexamethasone for nasal mucosa wound healing applications. The preclinical results from this research manuscript have shown that this composite can make a significant contribution to the biomedical fields. Moreover, the manuscript is well-written and organized. Henceforth, it can be considered for publication in pharmaceutics after the following minor revisions.
- Figure 1 must be cited in the content.
- In subsection 2.3., the quantities of reagents (i.e., DOPE, cholesterol, etc.) used for the fabrication of dexamethasone liposomes must be written.
- The TEM model number and country of manufacture must be mentioned.
Author Response
Comments 1: Figure 1 must be cited in the content.
Response 1: We appreciate the reviewer’s attention to this detail. Figure 1 is cited in the Introduction section to illustrate the design concept of the HYD-Lip/DXMS@HG (Page 2, Line 76).
Comments 2: In subsection 2.3., the quantities of reagents (i.e., DOPE, cholesterol, etc.) used for the fabrication of dexamethasone liposomes must be written.
Response 2: We have now explicitly included the specific quantities of DOPE (30 mg) and cholesterol (10mg) used in Section 2.3 (Page 4, Line 104-105).
Comments 3: The TEM model number and country of manufacture must be mentioned.
Response 3: To address this, we have updated the methodology in Section 2.3 to include the model number and country of manufacture: “To characterize the morphology and size of liposomes, TEM was performed using a TF20 microscope (FEI, USA) operated at 200 kV acceleration voltage.” (Page 4, Line 122).

Reviewer 3 Report
Comments and Suggestions for Authors
The current study aimed to formulate acylhydrazone-modified liposomes for dexamethasone delivery, the study is interesting, however the following should be addressed:
1-Please mention the amounts of DOPE, carboxylated cholesterol, and DSPE-HYD-PEG2000 and final drug concentration in liposomal formulae after hydration, also mention preparation method in a more detailed manner, how the evaporation under pressure was achieved
2- Add reference to preparation methods
3- Is the prepared liposomal formula results of a previous optimization study?
4-Mention methods for TEM, size, zeta-potential and EE% determination in more details
5-Mention final concentration of drug in the gel
6-Mention the maximum absorption wavelength of DXMS with a reference
7-Correct colony count in line 195.
8-What is the no of animal in each group, also specify the weight and sex.
9- The authors mention the drug in full terms in some sections and in abbreviated form in other sections.
10-The authors mention in Table 1 that there are 3 liposomal formulae, while in the methodology section the composition of these formulae not mentioned
11- Justify the following pharse. The UV spectrum showed that the liposome had an absorption peak at 235 nm, suggesting that dexamethasone was successfully loaded.
12-Please mention the impact of each formula composition on its characteristics as PS, ZP,EE%.
13- The authors mentioned that when low-concentration liposomes are incorporated into the hydrogel, the enhanced toughness and porosity enable the hydrogel to form more intimate contact with the substrate, thereby enhancing adhesion in lines 315& 316, are The authors study the impact of loading liposomal formulae in different concentrations on the characteristics of the gel
14-It was better to study release of liposomal dispersion and free drug-loaded gel at different pH to compare it with release of HYD-Lip/DXMS@HG.
15- In the release part, the drug was abbreviated to DEX?
Round 2
Reviewer 3 Report
Comments and Suggestions for Authors
The authors have made the requested corrections